# A Novel Concept of Electron–Hole Enhancement for Superjunction Reverse-Conducting Insulated Gate Bipolar Transistor with Electron-Blocking Layer

**DOI:** 10.3390/mi14030646

**Published:** 2023-03-12

**Authors:** Zhigang Wang, Chong Yang, Xiaobing Huang

**Affiliations:** 1School of Information Science and Technology, Southwest Jiao Tong University, Chengdu 610031, China; 2Qianghua Times (Chengdu) Technology Co., Ltd., Chengdu 610031, China

**Keywords:** reverse-conducting IGBT, snapback-free, thyristor, electron-blocking, superjunction IGBT, SOA, superjunction

## Abstract

A novel snapback-free superjunction reverse-conducting insulated gate bipolar transistor (SJ-RC-IGBT) is proposed and verified by simulation. In the SJ-RC-IGBT, the parasitic P/N/P/N structure as thyristor or Shockley diode demonstrates large conductivity due to an overabundance of carriers for reverse conduction. By preventing electrons from leaking across the N+ region at the collector side, the extra electron-blocking (EB) layer introduced in the SJ-RC-IGBT can dramatically enhance electron–hole pairs in the N/P-pillars. Hence, the SJ-RC-IGBT demonstrates a low on-state voltage (V_on_). In addition, snapback-free characteristics and a large safe operating area (SOA) are also achieved in the SJ-RC-IGBT. During the turn-off process, a significant amount of electrons are extracted by parasitic MOS across the EB layer at the collector side to decrease the turn-off loss (E_off_). According to the optimized results, the SJ-RC-IGBT with EB layer obtains an ultralow E_off_ of 3.9 mJ/cm^2^ at V_on_ = 1.38 V with 88% and 81% decreases, respectively, compared with the conventional reverse-conducting IGBT (CRC-IGBT) and superjunction IGBT (SJ-IGBT).

## 1. Introduction

The reverse-conducting insulated gate bipolar transistor (RC-IGBT) is a crucial component for packing in a single power module at a low package cost, and it plays vital roles in medium and high voltage inverters [1,2]. The RC-IGBT works in a forward and reverse state, necessitating [3,4,5,6]: (i) reverse conduction with a uniform current distribution; (ii) forward conduction with the lowest possible on-state voltage (V_on_) and without a snapback effect at the high carrier injection for conductive modulation; (iii) high-speed switching for the lowest possible turn-off power dissipation by accelerating carrier extraction. It is well known that the conventional RC-IGBT (CRC-IGBT) is achieved by anti-paralleling to a freewheeling diode (FWD) as an anode-short IGBT for reverse conduction. However, the CRC-IGBT easily results in a snapback phenomenon and a relatively high V_on_ at forward conduction and non-uniform current distribution at reverse conduction [3], which are, however, in conflict with the design requirements. The reverse-conducting capability and the snapback-free are desirable pursuits of the RC-IGBT design [7]. Although the snapback-free can be achieved by these techniques, such as a floating p-region in the trench collector [8] and discontinuous field-stop (FS) layer [9], these RC-IGBTs are at the expense of V_on_ and the forward conducting uniformity compared with conventional IGBTs. Therefore, in the RC-IGBT, an anti-parallel thyristor [10] or a Shockley diode [11] are introduced other than a FWD in the CRC-IGBT to realize both reverse conduction and suppress the snapback effect, which easily causes reverse snapback [7]. Moreover, the issue of current crowding is another challenge for obtaining a large safe operating area (SOA) in these RC-IGBTs. To eliminate the current-crowding problem, the edge-termination concept [12] is proposed and evaluated in the RC-IGBT to reduce injected holes into edge termination and avoid the recovery in the diode mode in lieu of the IGBT mode during the turn-off process. The current crowding is partially smoothed by the termination region, but the snapback effect is inevitable in the active region. To ensure snapback-free, a controllable collector trench gate RC-IGBT, which is controlled by an additional gate-driving signal [13], and an automatically controlled anode gate IGBT [14] are proposed, whereas the circuitry complexity or on-state resistance is also increased.

Further, the RC-IGBT needs to not only eliminate the snapback phenomenon but also account for the trade-off between V_on_ and turn-off loss (E_off_). To optimize this trade-off, the superjunction seems a good candidate for minimizing the E_off_ by unipolar and bipolar mode at a high or low doping concentration of the superjunction drift region [15]. The superjunction IGBT (SJ-IGBT) as an effective method to accelerate carrier extraction from the drift region for depletion has been proven to reduce E_off_ by simulations in [16], experiments in [17], and models in [18]. To the best of our knowledge, the suppression of the snapback effect in the RC-IGBT employing the superjunction is firstly realized by the anode-shorted structure at the sacrifice of V_on_ [19]. For reducing V_on_, a shorted-collector trench and carrier-storage (CS) layer introduced in the superjunction IGBT [20] can enhance carriers-injection for decreasing V_on_ and E_off_ but without a reverse-conducting capability. In addition, the “anode-side” superjunction IGBT helps effectively extract carriers and has potential for the RC-IGBT analogous to a MOS-controlled thyristor with a reduced snapback [21].

When compared to a standard RC-IGBT [19,21], a superjunction trench clustered insulated gate bipolar transistor (SJ-TCIGBT) with no reverse-conduction capability can simultaneously reduce V_on_ and E_off_. In this study, we also introduced a superjunction structure for RC-IGBTs [20]. In addition, the backside thinning method, which is added to traditional IGBTs as a thin n-base and a low-doped field-stop layer, can significantly minimize E_off_ [22]. In contrast to this feature, RC-IGBTs have a backside N+ doping area that reduces turn-off loss at the expense of substantial forward static losses. Therefore, a superjunction structure for RC-IGBTs includes a “double-side carrier storage layer” concept to improve conductance modulation in the drift zone and further reduce the V_on_ compared with the RC-IGBTs in [19,20,21].

In this paper, the state-of-the-art superjunction reverse-conducting IGBT (SJ-RC-IGBT) for superior V_on_-E_off_ trade-off is proposed and verified by simulation. A feature is that an additional trench collector is inserted into the N/P-pillars for separating the N+ and P+ collector regions. In the reverse conduction, the parasitic P/N/P/N thyristor in the SJ-RC-IGBT is triggered on for a large conductivity to latch up other parasitic BJTs. Both of the N/P-pillars can offer a conduction path to uniform the space charges distribution. In the forward conduction, the V_on_ is reduced by a large amount of excess hole injected from P+ collector region into the N-pillar and its carrier-storage effect is enhanced by the CS layer. In addition, the electron-blocking (EB) layer can also enhance the carrier-storage effect, which restricts the electrons in the N/P-pillars leaking via the bypath of this trench collector into the N+ collector region for achieving a low V_on_. Highlighting that the carrier-storage effect is weakened by anode-shorted N+ region in the collector is correspondingly compensated by the EB layer to eliminate the snapback effect. Nevertheless, the inherent large account of excess carriers can be extracted rapidly during the turn-off process. E_off_ is dramatically reduced by this extraction path generated by the N+ collector region, EB layer, and trench collector.

In Section 2, the forward- and reverse-conducting mechanisms of the SJ-RC-IGBT are investigated and revealed. In Section 3, the optimized results are discussed for the SJ-RC-IGBT performance. At the forward breakdown voltage of 1540 V, a small V_on_ of 1.27 V is obtained much lower than that of the CRC-IGBT at E_off_ = 3.65 mJ/cm^2^.

## 2. Device Structure and Conduction Mechanism

### 2.1. Device Structure Profile

Figure 1a–c show the schematic views of the SJ-RC-IGBT, SJ-IGBT and CRC-IGBT, respectively. The dimensions are shown in the figure. If it has no special specification, the key doping concentrations are N_pillar_ = 3 × 10^15^ cm^−3^-doping concentration of the N/P-pillars, N_CS_ = 1 × 10^16^ cm^−3^-doping concentration of the CS layer, and N_FS_ = 2 × 10^16^ cm^−3^-doping concentration of the FS layer. In the SJ-RC-IGBT, the doping concentration of the EB layer (N_EB_) is 2 × 10^16^ cm^−3^. In the CRC-IGBT, the doping concentration of N-drift (N_d_) is 1 × 10^13^ cm^−3^. Two-dimensional simulator MEDICI have been carried out, and the physical models—CONMOB (carrier mobility model dependence on doping concentration), FLDMOB (carrier mobility model dependence on high electric field), SRFMOB2 (enhanced surface mobility model), CONSRH (Shockley–Read–Hall recombination model), AUGER (Auger recombination model), BGN (Slotboom bandgap narrowing model), and IMPACT.I (impact ionization model)—are included [22]. The simulation is verified with data and calibrated [23]. The carrier lifetime (τ) is set at 1 µs [24].

Compared with the SJ-IGBT and CRC-IGBT, the main feature is that an EB layer at the bottom of the P-pillar and an additional trench collector inserted into the N-pillar and P-pillar to separate the N+/P+ collector regions are implanted in the SJ-RC-IGBT. At the emitter side, the injected holes in the N/P-pillars are blocked by the CS layer, and at the collector side, the injected electrons are blocked by the EB layer. Therefore, electron–hole pairs in the N/P-pillars are further improved by the CS layer and EB layer higher than the SJ-RC-IGBT without the EB layer. In the SJ-RC-IGBT, the EB layer and CS layer help the reduction of V_on_. During the turn-off process, the trench collector can speed up extraction of the excess carriers from the N/P-pillars for a low E_off_. Otherwise, if the CS layer is introduced in the SJ-IGBT, the optimization of the V_on_-E_off_ trade-off is also achieved but without the reverse-conducting capability, as given in [22].

In order to illustrate the carrier-storage effect of the EB layer and CS layer, Figure 2a shows the schematic energy-band views and carrier distributions at V_G_ = 15 V and V_CE_ = 1 V. The doping concentration of the EB layer increasing from 1.0 × 10^16^ cm^−3^ to 2.0 × 10^16^ cm^−3^ can enhance the electron concentration along line BB’ from ~1.5 × 10^15^ cm^−3^ to ~5.8 × 10^15^ cm^−3^ in the P-pillar and from ~5.5 × 10^15^ cm^−3^ to ~9.8 × 10^15^ cm^−3^ in the N-pillar, as shown in Figure 2b. The structure without the EB layer means that the corresponding region is filled by the P-pillar. It is obviously found that the EB layer as electron barrier can enhance the carrier-storage effect in the N/P-pillars, but in the absence of the EB layer, the hole-storage effect by the CS layer is weakened.

### 2.2. Reverse- and Forward-Conducting Mechanism

Figure 3 and Figure 4 show the reverse/forward-conducting mechanisms of the SJ-RC-IGBT, the corresponding equivalent circuits, and the current distributions under different conduction current density. The parasitic Bipolar Junction Transistors (BJTs) are marked in Figure 3a and Figure 4a.

At reverse conduction, holes are injected from the P+ region at the emitter side into the N/P-pillars and electrons are injected from the N+ region at the collector side, as shown in Figure 3a. In the beginning, BJTs B_7_ and B_8_ are turned on as the first thyristor, then BJTs B_6_ and B_11_ are turned on as the second thyristor. With an increase of the reverse current, the voltage drop across the junction of the P-body/CS layer is larger than ~0.7 V, with BJTs B_9_ and B_10_ triggered on. At last, BJT B_5_ is turned on by the effective base current at a positive biased emitter. The equivalent circuit of the SJ-RC-IGBT at reverse conduction is illustrated in Figure 3b.

Figure 3c–f show the distributions of current at reverse conduction. At 10 mA/cm^2^, holes are injected from the emitter region and electrons are injected from the collector region of the SJ-RC-IGBT as two serial P/N junctions. Because of the narrow CS layer, BJTs B_7_ and B_8_ work as a thyristor with a low voltage drop. With an increase of reverse bias of the emitter, the conduction current is increased from 40 mA/cm^2^ to 70 mA/cm^2^ for BJTs B_6_, B_9_, B_10_, and B_11_ on and from 70 mA/cm^2^ to 500 mA/cm^2^ for BJT B_5_ on. Eventually, at a high reverse current condition, the N/P-pillars become high conductivity regions.

At forward conduction, the parasitic MOSFETs (M_1_, M_2_, and M_3_) around the gate are firstly turned on. Electrons from the N+ emitter region and holes from the P+ collector region are injected into the N/P-pillars, as shown in Figure 4a. The forward equivalent circuit is illustrated in Figure 4b. The injected electrons into the N-pillar trigger on BJTs B_1_ and B_2_. With the forward current increasing, the junction of the P-pillar/CS layer will be forward biased sufficiently large enough to inject minority carriers from the P-pillar to the CS layer, and then BJT B_4_ is triggered on. Since the base region of BJT B_4_ is formed by a narrow CS layer with a relatively high doping concentration, BJT B_4_ works as a low gain BJT. As forward conduction increases, the junction of the P-pillar/CS layer is further positively biased. The parasitic BJT B_3_ is turned on. There is a weak conduction channel for electrons in M_4_ (parasitic MOSFET around the trench collector) to offer a path for electron extraction when the SJ-RC-IGBT is turned off.

Figure 4c–f show the distributions of current at forward conduction. At low forward conduction (10 mA/cm^2^), electrons injected from MOSFET M_2_ as unipolar current is intended for the base of BJT B_1_. Up to 50 mA/cm^2^, holes injected from the P+ collector region and electrons injected from MOSFETs M_3_ and M_1_ can activate BJTs B_1_ and B_2_. When the conduction current increases to 100 mA/cm^2^, BJT B_4_ works at on-state. At last, the N+ collector region potential is high enough for triggering BJT B_3_ on at 500 mA/cm^2^. There are two conduction channels with a low barrier for electrons from MOSFET M_1_ to MOSFET M_4_ around the trench collector.

## 3. Results and Discussion

### 3.1. Conduction Characteristics

Figure 5 shows the forward/reverse I-V curves of the SJ-RC-IGBT, SJ-IGBT, and CRC-IGBT. From Figure 5a, it is obviously found that the SJ-IGBT and SJ-RC-IGBT turn on with snapback-free, but the CRC-IGBT has a snapback phenomenon at L_P_ = 170 μm and 200 μm. Due to the inhibition of unipolar conduction provided by the N+ collector area, the snapback effect in the CRC-IGBT can be fully removed when L_P_ increases to 270 μm. As shown in the inset of Figure 5a, the forward conduction exceeds that of the SJ-IGBT after I_CE_ of the SJ-RC-IGBT exceeds 33.1 A/cm^2^. The minimum voltage drop (1.24 V) at I_CE_ = 100 A/cm^2^ is also obtained in the SJ-RC-IGBT. Figure 5b shows the reverse I-V curves of the SJ-RC-IGBT and CRC-IGBT. Although the reverse voltage drop of the CRC-IGBT can be decreased with the reduction of L_P_, this is a contradiction as an elimination of the snapback phenomenon. However, the SJ-RC-IGBT demonstrates superior performance: a small reverse voltage drop and a uniform distribution of current. The thyristor is prone to snapback during the conduction process due to the positive feedback of the P/N/P/N structure needing some conditions, with the main factor being τ. The positive feedback cannot be formed if the τ is too small, as the thyristor will withstand voltage until the minority carriers can pass through the base region, then the snapback will occur. In [11], the RC-IGBT with Shockley diode (SH-RC-IGBT) has a snapback-free at a reverse voltage drop of 0.95 V. Further lowering the reverse voltage drop in SH-RC-IGBT makes it challenging to activate the parasitic thyristor without snapback. Hence, the RC-IGBT with Shockley diode (SH-RC-IGBT) has a snapback phenomenon even though it has the bigger τ with lower reverse voltage drop, as shown is Figure 5c, but it is snapback-free for the SJ-RC-IGBT. This is due to the electron being injected into the N-pillar through the boundary of the trench collector.

### 3.2. Safe Operating Area

Conventionally, the instability of the IGBT is prone to trigger on parasitic BJTs, giving rise to a non-uniform current distribution at quasi-saturation conduction and high voltage bias. As shown in Figure 6a, the I-V curves of the CRC-IGBT snapback below 800 V much lower than that of the SJ-IGBT, as shown in Figure 6b, and the SJ-RC-IGBT, as shown in Figure 6c. This is mainly because the non-uniform current regenerates high space charge modulation in the absence of a uniform electric field. In the SJ-IGBT and SJ-RC-IGBT, the N/P-pillars can offer a uniform conduction path without crowding space charges and avoiding immature breakdown. Although the saturation current density of the SJ-RC-IGBT is higher than the CRC-IGBT, the SOA of the SJ-RC-IGBT is continuously improved, due to uniform current regions sustained by the N/P-pillars.

### 3.3. BV and V_on_ Optimization

Figure 7a shows the breakdown voltage (BV) versus N_EB_ and N_CS_. In the range of N_EB_ = 1.2 × 10^16^ cm^−3^–2.4 × 10^16^ cm^−3^, BV has a small variation at a certain N_CS_. Figure 7b shows V_on_ versus N_EB_ and N_CS_. During forward conduction, V_on_ is influenced by the ambipolar effect. The increase of N_CS_ leads to a high hole potential barrier near the P-body to restrict hole extraction. Meanwhile, the increase of N_EB_ results in raising the electron potential barrier to enhance hole injection in the N-pillar for achieving high conductive modulation. Hence, both an N_CS_ and N_EB_ increase can reduce V_on_.

### 3.4. Turn-Off Characteristics

Figure 8a shows the turn-off process of the SJ-RC-IGBT, SJ-IGBT, and CRC-IGBT. In the SJ-RC-IGBT, I_CE_ falling from 90 A/cm^2^ to 10 A/cm^2^ needs 46 ns by a huge reduction of 95.8% compared with the CRC-IGBT and 86.7% compared with the SJ-IGBT. The high performance of the turn-off process is generally attributed to two aspects—(i) low injection of excess carriers, as shown in Figure 8b, and (ii) extracting channels, as implied in Figure 8c. (i) In the SJ-RC-IGBT, the injection of holes is at a low level, but the conductive modulation is at high level in contrast with the SJ-IGBT and CRC-IGBT. (ii) At t_1_, a high density of electron–hole pairs exists at all of the CS layer, N/P-pillars, and EB layer. Although from t_2_ to t_3_, the electron and hole in the main regions of the CS layer, N/P-pillars, and EB layer are remarkably reduced, a relatively high hole density remains along the left side of the gate in the CS layer and electron channel along the trench collector in the EB layer.

### 3.5. Trade-Off between Forward Conduction and Turn-Off Loss

Figure 9 shows the trade-off between E_off_ and V_on_ at I_CE_ = 100 A/cm^2^. With an increase of N_EB_ from 1.0 × 10^16^ cm^−3^ to 2.0 × 10^16^ cm^−3^, V_on_ of the SJ-RC-IGBT is reduced from 1.38 V to 1.27 V at E_off_ = 4.25 mJ/cm^2^, as shown in Figure 9. It can be seen that the SJ-RC-IGBT with EB layer (N_EB_ = 1.0 × 10^16^ cm^−3^) has V_on_ of 1.38 V at E_off_ = 4.25 mJ/cm^2^, which is 17% lower than the SJ-IGBT, as shown in Figure 9. It is worth emphasizing that the EB layer in the SJ-RC-IGBT can enhance the conductivity in the N/P-pillars for gaining a lower V_on_ due to restricting electrons leaking via the P-pillar into the N+ collector region, as the effect of the CS layer for blocking holes. Nevertheless, if the SJ-RC-IGBT is without the EB layer, a high speed of turn-off switching is easy to obtain in the SJ-RC-IGBT from green dots, as shown in Figure 9, but it inevitably results in improving V_on_. In brief, the SJ-RC-IGBT with EB layer exhibits superior E_off_-V_on_ trade-off over the FP-RC-IGBT [8], SJ-CSTBT [22], SH-RC-IGBT [11], SJ-IGBT, and CRC-IGBT.

## 4. Key Fabrication Process of the SJ-RC-IGBT

Figure 10 shows the key fabrication process of the SJ-RC-IGBT. First, a deep trench is formed by reactive ion etching, and the deep trench can achieve a fairly large aspect ratio, as shown in Figure 10a. Figure 10b shows the trench refilling by anisotropic epitaxial growth; after epitaxy, the flat surface is achieved by chemical mechanical polishing. The surface defects were then repaired by sacrificing oxidation. The final surface is shown in Figure 10b. Before forming the MOS part, the CS layer and P-body are formed by epitaxial growth, as shown in Figure 10c. The rest of the facade is formed by standard trench gate process, with the final configuration of the emitter region and trench gate shown in Figure 10d. The backside process is essential in the design of the RC-IGBT. First, through the back thinning to achieve the required width of the drift region, as shown in Figure 10e, then the FS layer, P+ collector, EB layer, and N+ collector are formed by backside implantation, as shown in Figure 10f,g. Last, the trench collector and electrode process on the back is similar to that on the front. Through the above process flow, the SJ-RC-IGBT finally forms the structure shown in Figure 10h.

## 5. Conclusions

A SJ-RC-IGBT with EB layer is proposed and investigated for revealing its complex conduction mechanism at the forward and reverse conditions. This innovative mechanism aids optimizing the trade-off between E_off_ and V_on_. In comparison with the CRC-IGBT, the SJ-IGBT and SJ-RC-IGBT without EB layer and E_off_ and V_on_ in SJ-RC-IGBT with EB layer demonstrate superior advantages. Moreover, the N/P-pillars to deliver a uniform distribution of space charges without compromising the SOA at high forward conduction has considerably increased the SOA of the SJ-RC-IGBT.

## Figures and Tables

**Figure 1 micromachines-14-00646-f001:**
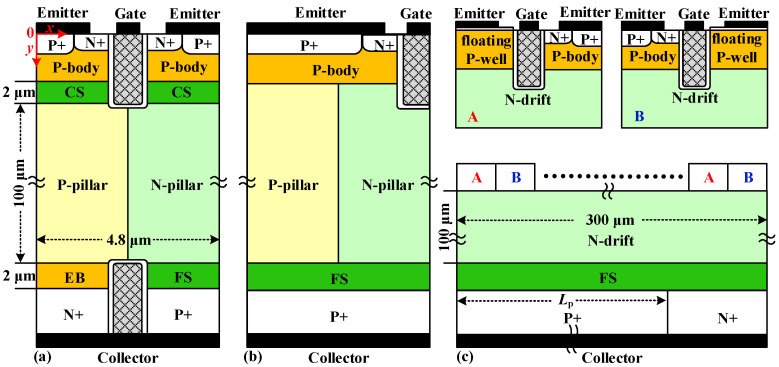
Schematic cross-sections of (**a**) SJ-RC-IGBT, (**b**) SJ-IGBT, and (**c**) CRC-IGBT. The cell-unit width of SJ-RC-IGBT and SJ-IGBT is 4.8 μm and of CRC-IGBT is 300 μm.

**Figure 2 micromachines-14-00646-f002:**
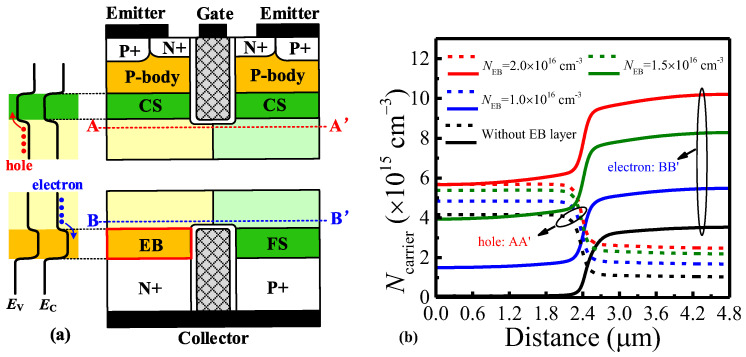
The effect of N_EB_ on carrier distribution. (**a**) Diagram of carrier-storage effect along cut lines position. (**b**) The influence of N_EB_ on hole concentration in the vicinity of CS layer and electron concentration in the vicinity of EB layer at the condition of V_G_ = 15 V and V_CE_ = 1 V.

**Figure 3 micromachines-14-00646-f003:**
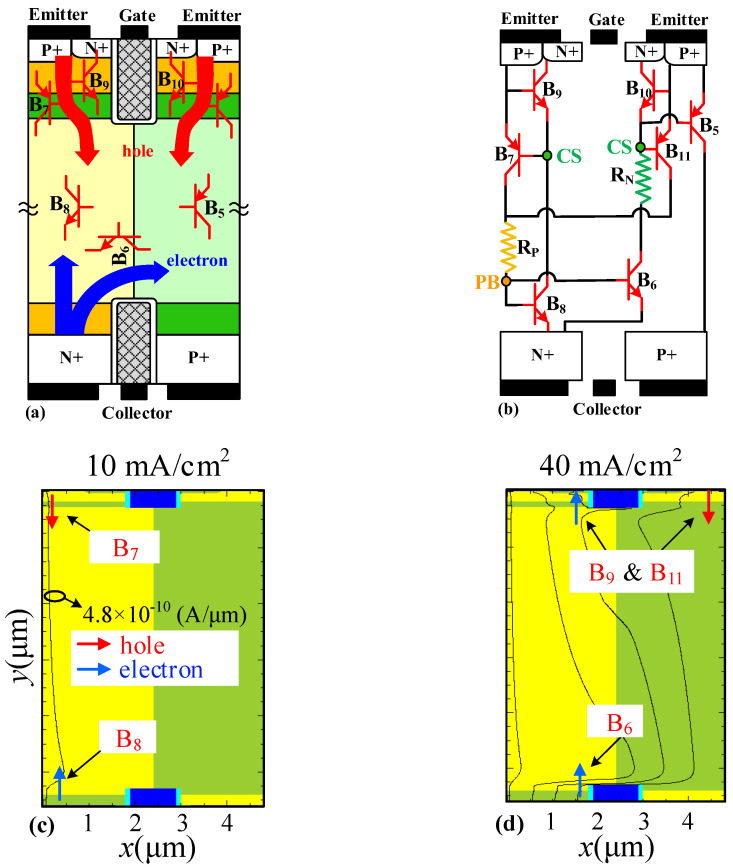
The reverse conduction and equivalent circuit of SJ-RC-IGBT. (**a**) Reverse-conducting mechanism. (**b**) Equivalent circuit at reverse conduction. Reverse-conducting current distribution of SJ-RC-IGBT at (**c**) 10 mA/cm^2^, (**d**) 40 mA/cm^2^, (**e**) 70 mA/cm^2^, and (**f**) 500 mA/cm^2^. The bias condition of reverse conduction is that the collector and gate are grounded, and the emitter is positively biased. All the contours of current density are 4.8 × 10^−10^ A/μm. Blue arrows represent electron injection and red arrows represent hole injection.

**Figure 4 micromachines-14-00646-f004:**
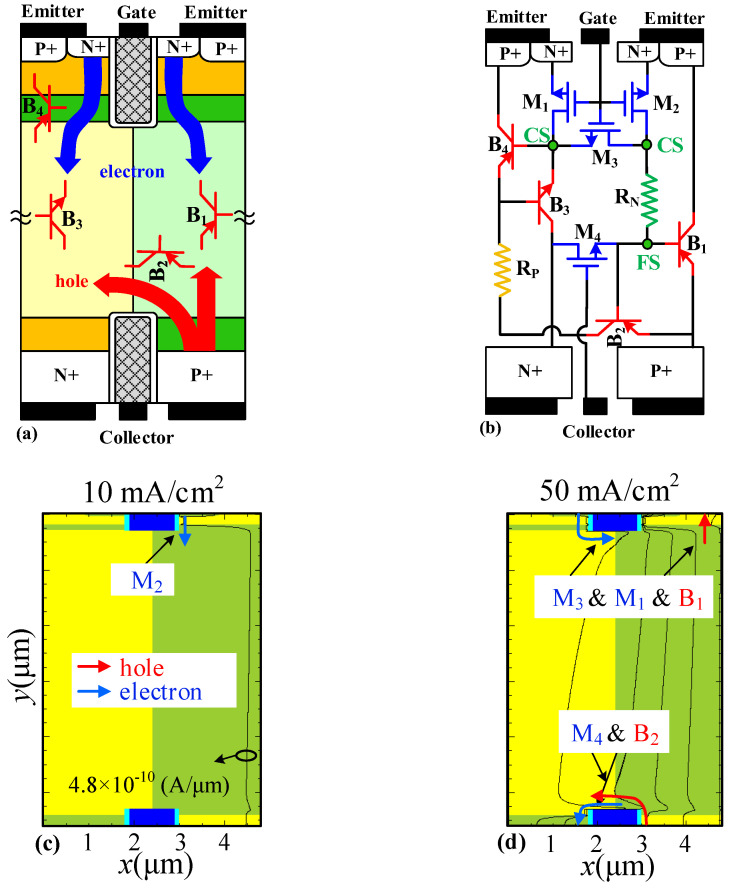
The forward conduction and equivalent circuit of SJ-RC-IGBT. (**a**) Forward-conducting mechanism. (**b**) Equivalent circuit at forward conduction. Forward-conducting current distribution of SJ-RC-IGBT at (**c**) 10 mA/cm^2^, (**d**) 50 mA/cm^2^, (**e**) 100 mA/cm^2^, and (**f**) 500 mA/cm^2^. The bias condition of forward conduction is that the gate and collector are positively biased, and the emitter is grounded. All the contours of current density are 4.8 × 10^−10^ A/μm. Blue arrows represent electron injection and red arrows represent hole injection.

**Figure 5 micromachines-14-00646-f005:**
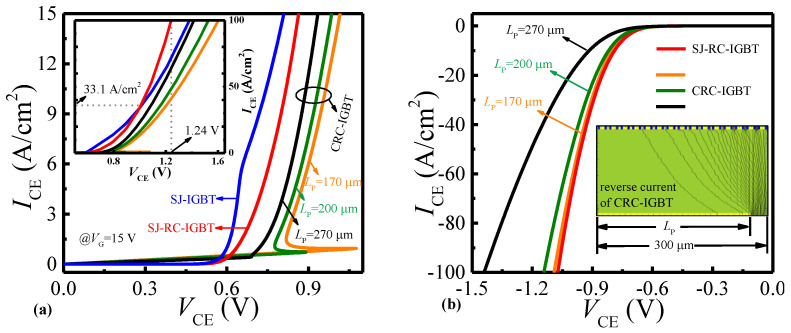
(**a**) Forward conduction of I_CE_ versus V_CE_ and (**b**) reverse conduction of I_CE_ versus V_CE_. In CRC-IGBT, Lp (the length of P+ collector region) is changed at forward and reverse conduction. (**c**) Reverse snapback curves. FP-RC-IGBT is referenced from [8].

**Figure 6 micromachines-14-00646-f006:**
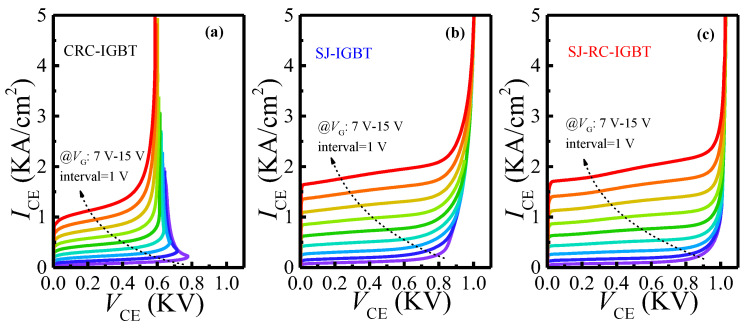
I-V characteristics of three IGBTs with a ramp-up V_G_ from 7 V to 15 V(The lines from blue-purple to red indicate increase in V_G_ from 1 V to 5 V with 1 V interval.): (**a**) I-V characteristics of CRC-IGBT, (**b**) I-V characteristics of SJ-IGBT, and (**c**) I-V characteristics of SJ-RC-IGBT.

**Figure 7 micromachines-14-00646-f007:**
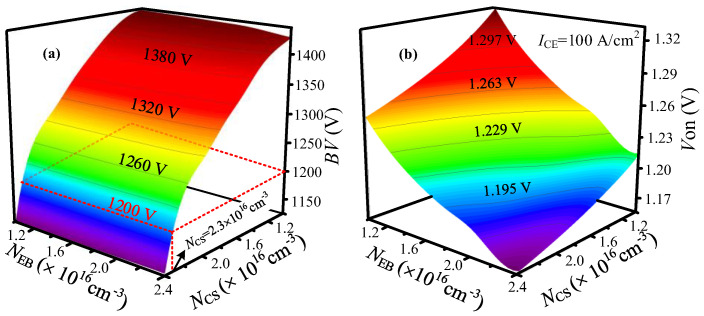
BV and V_on_ dependence on doping concentration in SJ-RC-IGBT. (**a**) Effect of N_CS_ and N_EB_ on BV. (**b**) Effect of N_CS_ and N_EB_ on V_on_.

**Figure 8 micromachines-14-00646-f008:**
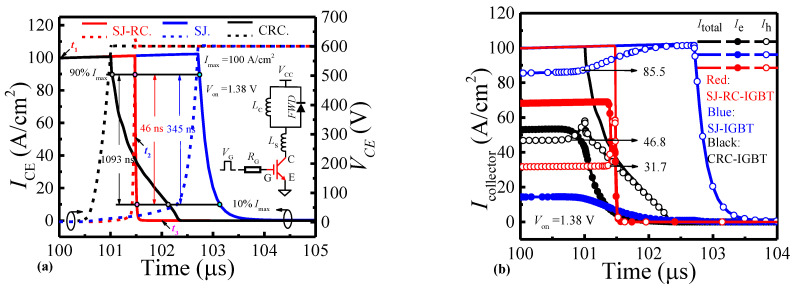
(**a**) Turn-off waveforms of SJ-RC-IGBT, SJ-IGBT, and CRC-IGBT. The inset shows the simulation circuit with R_G_ = 10 Ω and Ls = 5 nH. (**b**) Electron current (I_e_), hole current (I_h_), and total current (I_total_) at the collector of SJ-RC-IGBT, SJ-IGBT, and CRC-IGBT during turn-off. (**c**) Electron/hole concentrations in CS layer, N/P-pillars, and EB/FS layer of SJ-RC-IGBT. Hollow and solid represent hole and electron, respectively.

**Figure 9 micromachines-14-00646-f009:**
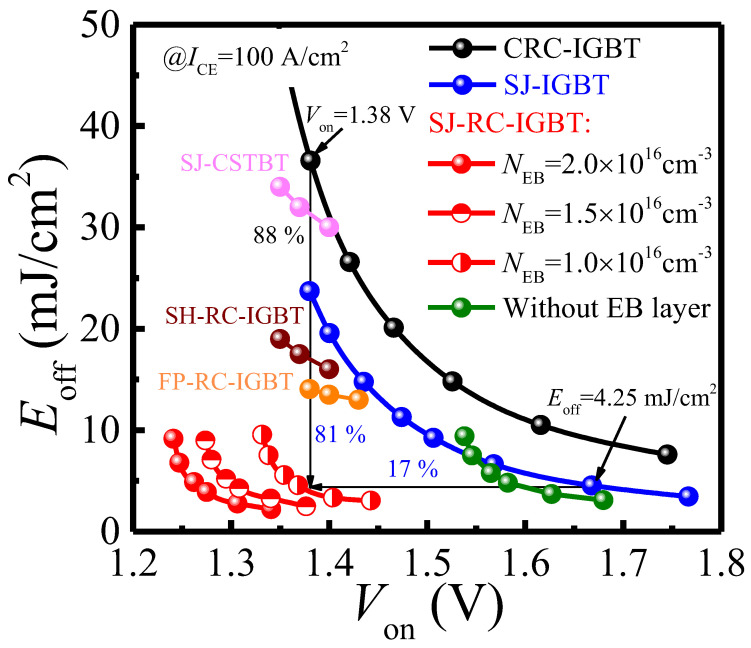
E_off_-V_on_ relationship of SJ-RC-IGBT, SJ-IGBT, and CRC-IGBT. E_off_-V_on_ relationship at I_CE_ = 100 A/cm^2^. FP-RC-IGBT is referenced from [8]. SH-RC-IGBT is referenced from [11]. SJ-CSTBT is referenced from [22].

**Figure 10 micromachines-14-00646-f010:**
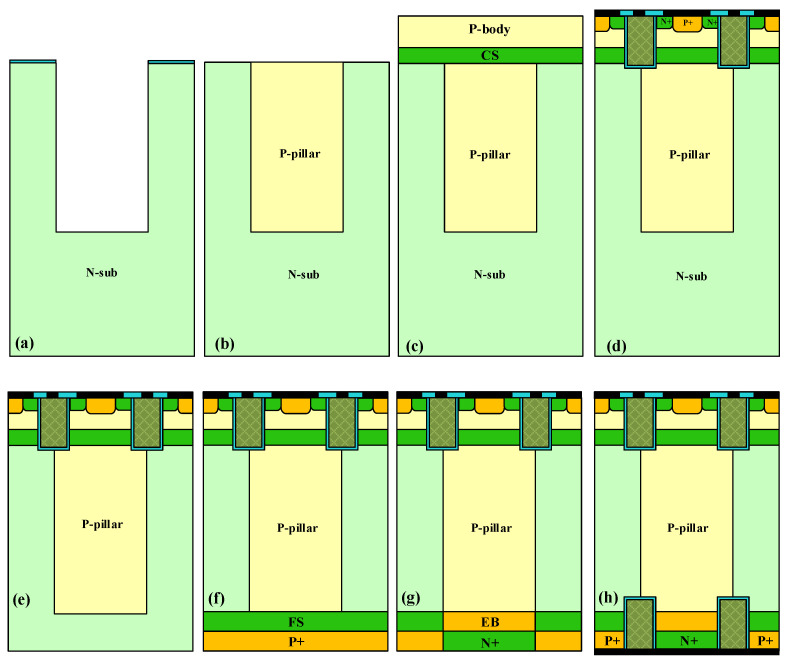
The key fabrication process steps of the SJ-RC-IGBT. (**a**) Deep trench etching based on reactive ion etching, (**b**) trench refilling by anisotropic epitaxial growth, (**c**) epitaxial growth of CS layer and P-body, (**d**) forming trench MOS structure and electrode, (**e**) back thinning, (**f**) forming FS layer and P+ collector by backside implantation, (**g**) forming EB layer and N+ collector by backside implantation, and (**h**) forming backside trench and electrode.

## Data Availability

Not applicable.

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
