# Peer review of "A Novel Concept of Electron–Hole Enhancement for Superjunction Reverse-Conducting Insulated Gate Bipolar Transistor with Electron-Blocking Layer"

_micromachines, 2023, doi:10.3390/mi14030646_

Round 1
Reviewer 1 Report
1. the density of electrons in line 120 and 121 in not consistent with Fig. 2(b), please check again. in addtion, for the case without EB layer, what is in this region? FS layer? please describe clearly.
2. In fig. 3 and 4, the author gave the current density distribution, suggest to distinguish the hole current and the electron current, is the current in n pillar is mainly electron current?
3. the colletor electrode will endure high voltage, please the author to describe how to form the trench collector, and is it filled firstly by dielectrics and then by poly-Si or metal, here the electical field should be very high, how to solve this problem?
Author Response
Response to Reviewer 1 Comments
Point 1: The density of electrons in line 120 and 121 in not consistent with Fig. 2(b), please check again. in addtion, for the case without EB layer, what is in this region? FS layer? please describe clearly.
Response 1:
Thanks to the reviewer for pointing out that our description is not detailed.
We have rechecked the density of electrons in line 120 and 121 is correct, which is in consistent with Fig. 2(b).
For clear description, the sentence is added in revised manuscript as
“The structure without EB layer means that the corresponding region is filled by P-pillar.”
Point 2: In fig. 3 and 4, the author gave the current density distribution, suggest to distinguish the hole current and the electron current, is the current in n pillar is mainly electron current?
Response 2: Thanks to the reviewer for pointing out this.
In Fig. 3 and 4, the author gave the current density distribution. The current density is total density. In n-pillar, the current during on state is not only electron current due to the large conductivity effect. Hence, at both of forward and reverse state, the current is coeffect of the hole current and electron.
Point 3: The colletor electrode will endure high voltage, please the author to describe how to form the trench collector, and is it filled firstly by dielectrics and then by poly-Si or metal, here the electical field should be very high, how to solve this problem?
Response 3: Thanks to the reviewer for pointing out this.
At collector is always connected with high voltage except reverse bias. The differential potential between trench collector and the bottom of the drift region is not large. Meanwhile, the dielectric material has large critical electric field to make it possible to sustain the voltage. So the fabrication of this trench collector is filled dielectric and then by poly-Si as your suggestion.
The key fabrication process of the SJ-RC-IGBT is also added in revised manuscript as
Fig. 10 shows the key fabrication process of the SJ-RC-IGBT. First, deep trench is formed by reactive ion etching, the deep trench can achieve a fairly large aspect ratio as shown in Fig. 10(a). Fig. 10(b) shows the trench refilling by anisotropic epitaxial growth, after epitaxy, the flat surface is achieved by chemical mechanical polishing. Then, surface defects were repaired by sacrificing oxidation. The final surface is shown in Fig. 10(b). Before forming the MOS part, CS layer and P-body are formed by epitaxial growth as shown in Fig. 10(c). The rest of the facade is formed by standard trench gate process, the final configuration of the emitter region and trench gate is shown in Fig. 10(d). Backside process is essential in the design of RC-IGBT. First through the back thinning to achieve the required width of the drift region as shown in Fig. 10(e), then FS layer, P+ collector, EB layer and N+ collector are formed by backside implantation as shown in Fig. 10(f) and Fig. 10(g), the last, trench collector and electrode process on the back is similar to that on the front. Through the above process flow, the SJ-RC-IGBT finally forms the structure shown in the Fig. 10(h).
Fig. 10. The key fabrication process steps of the SJ-RC-IGBT. (a) Deep trench etching based on reactive ion etching, (b) trench refilling by anisotropic epitaxial growth, (c) epitaxial growth of CS layer and P-body, (d) forming trench MOS structure and electrode, (e) back thinning, (f) forming FS layer and P+ collector by backside implantation, (g) forming EB layer and N+ collector by backside implantation, (h) forming backside trench and electrode.

Reviewer 2 Report
The authors have submitted work on the effect of electron-blocking layer in an Insulated gate bipolar transistor. The presented results are detailed and clearly explained. I would recommend the work given that the authors must clarify a few minor concerns. Is the proposed simulation methodology calibrated with experimental data to validate the results? If yes, what parameters are used to tweak the characteristics to match the experimental data? Explain the fabrication process of the proposed structure. Even for thyristor-based operation, the effect of impact ionization is really low. Kindly check the proposed device with and without the impact ionization model. Are any of the selberherr parameters changed while using the II? Comment on the scalability of the device. How do the authors suggest controlling the carrier lifetime?
Author Response
Response to Reviewer 2 Comments
Point 1: The authors have submitted work on the effect of electron-blocking layer in an Insulated gate bipolar transistor. The presented results are detailed and clearly explained. I would recommend the work given that the authors must clarify a few minor concerns. Is the proposed simulation methodology calibrated with experimental data to validate the results? If yes, what parameters are used to tweak the characteristics to match the experimental data? Explain the fabrication process of the proposed structure. Even for thyristor-based operation, the effect of impact ionization is really low. Kindly check the proposed device with and without the impact ionization model. Are any of the selberherr parameters changed while using the II? Comment on the scalability of the device. How do the authors suggest controlling the carrier lifetime?
Response 1: Please provide your response for Point 1. (in red)
Thanks to the reviewer for pointing out that fabrication process is not included. We have added a detailed fabrication of SJ-RC-IGBT in revised manuscript as
Fig. 10 shows the key fabrication process of the SJ-RC-IGBT. First, deep trench is formed by reactive ion etching, the deep trench can achieve a fairly large aspect ratio as shown in Fig. 10(a). Fig. 10(b) shows the trench refilling by anisotropic epitaxial growth, after epitaxy, the flat surface is achieved by chemical mechanical polishing. Then, surface defects were repaired by sacrificing oxidation. The final surface is shown in Fig. 10(b). Before forming the MOS part, CS layer and P-body are formed by epitaxial growth as shown in Fig. 10(c). The rest of the facade is formed by standard trench gate process, the final configuration of the emitter region and trench gate is shown in Fig. 10(d). Backside process is essential in the design of RC-IGBT. First through the back thinning to achieve the required width of the drift region as shown in Fig. 10(e), then FS layer, P+ collector, EB layer and N+ collector are formed by backside implantation as shown in Fig. 10(f) and Fig. 10(g), the last, trench collector and electrode process on the back is similar to that on the front. Through the above process flow, the SJ-RC-IGBT finally forms the structure shown in the Fig. 10(h).
Fig. 10. The key fabrication process steps of the SJ-RC-IGBT. (a) Deep trench etching based on reactive ion etching, (b) trench refilling by anisotropic epitaxial growth, (c) epitaxial growth of CS layer and P-body, (d) forming trench MOS structure and electrode, (e) back thinning, (f) forming FS layer and P+ collector by backside implantation, (g) forming EB layer and N+ collector by backside implantation, (h) forming backside trench and electrode.
As we already indicated, simulation software has been able to generate simulation results from reference that are reasonably realistic.
- Luther-King, E. M. S. Narayanan, L. Coulbeck, A. Crane, and R. Dudley, "Comparison of Trench Gate IGBT and CIGBT Devices for Increasing the Power Density From High Power Modules," Ieee Transactions on Power Electronics, vol. 25, no. 3, pp. 583-591, Mar. 2010, doi: 10.1109/Tpel.2009.2030327.
In this literature, it analyzes the device characteristics by means of simulation to verify the experiments. Figure 3 from this reference shows the comparison between simulation and experiments of the turn-off transient in this literature, and it can be observed that the simulation results and experiments match quite well. All design in our manuscript is also verified by those results. So the sentence “The simulation is verified with data and calibrated as [23]. The carrier lifetime (τ) is set at 1 µs [24].” is added in our revised manuscript.
For thyristor-based operation, the effect of impact ionization is really low. We have carefully checked this model and in the case with and without the impact ionization model. The current characteristics at on-state has not changed with or without this model. The main reason for this is that the conductivity modulation effect is consist of large non-equilibrium carrier. In order to extend the scalability of IGBT and applicability of the simulations, we have used the simulated carrier lifetime settings which is the same as in the literature. The corresponding sentence is at line 113, as “The carrier lifetime (τ) is set at 1 µs.”

Reviewer 3 Report
This paper proposes the RC-IGBT based on the superjunction structure. I have the following comments.
1. The simulation should be calibrated based on the data from real devices.
2. The authors should mention the novelty of their study compared to ref. [19], [20], and [21].
3. Is there any expected difference when the simulation is expanded to 3D?
4. In Figure 5(c), it seems that the ref. [11] exhibited the snapback characteristic. However, the title of ref. [11] is "A novel snapback-free reverse conducting IGBT ~" and the ref. [11] showed the snapback-free characteristic in SH-RC-IGBT. The authors should revise/add some data/description.
Author Response
Response to Reviewer 3 Comments
Point 1: The simulation should be calibrated based on the data from real devices.
Response 1:
Thanks very much for your careful reading of our manuscript for this good comment. We have modified the detailed verification of simulation to be clearer and more acceptable. As we mentioned earlier, simulation software has been able to achieve fairly accurate simulation results from reference.
- Luther-King, E. M. S. Narayanan, L. Coulbeck, A. Crane, and R. Dudley, "Comparison of Trench Gate IGBT and CIGBT Devices for Increasing the Power Density From High Power Modules," Ieee Transactions on Power Electronics, vol. 25, no. 3, pp. 583-591, Mar. 2010, doi: 10.1109/Tpel.2009.2030327.
In this literature, it analyzes the device characteristics by means of simulation to verify the experiments. Figure 3 from this reference shows the comparison between simulation and experiments of the turn-off transient in this literature, and it can be observed that the simulation results and experiments match quite well. All design in our manuscript is also verified by those results. So the sentence “The carrier lifetime (τ) is set at 1 µs. The simulation is verified with data and calibrated as [23].” is added in our revised manuscript.
Point 2: The authors should mention the novelty of their study compared to ref. [19], [20], and [21].
Response 2:
Thanks to the reviewer for suggestion that our description. We have added a detailed comparison.
To the best of our knowledge, the suppression of snapback effect in RC-IGBT employing superjuntion is firstly realized by the anode-shorted structure at the sacrifice of Von in [19]. For reducing Von, a shorted-collector trench and carrier-storage (CS) layer introduced in superjunction IGBT [20] can enhance carriers-injection for decreasing Von and Eoff but without reverse-conducting capability. In addition, the “anode-side” superjunction IGBT helps effectively extract carriers and has potential for RC-IGBT analogous to MOS-controlled thyristor with a reduced snapback [21].
[19]Antoniou, M.; Udrea, F.; Bauer, F.; Nistor, I. A new way to alleviate the RC IGBT snapback phenomenon: the superjunction solution. In Proc. Int. Symp. Power Semiconductor Devices ICs (ISPSD), Japan, August 2010.
[20]Zhou, K.; Luo, X. R.; Huang, L. H.; Liu, Q.; Sun, T.; Li, Z. J.; Zhang, B. An ultralow loss superjunction reverse blocking insulated-gate bipolar transistor with shorted-collector trench. IEEE Electron Device Lett. 2016, 37, 1462-1465.
[21]Antoniou, M.; Lophitis, N.; Udrea, F.; Bauer, F.; Vemulapati, U. R.; Badstuebner, U. On the investigation of the “Anode Side” superJunction IGBT design concept. IEEE Electron Device Lett. 2017, 38, 1063-1066.
In our revised manuscript, the following paragraph is added as
“When compared to a standard RC-IGBT [19,21], a superjunction trench clustered insulated gate bipolar transistor (SJ-TCIGBT) with no reverse-conduction capability can simultaneously reduce Von and Eoff. In this study, we also introduced a superjunction structure for RC-IGBTs [20]. In addition, the back-side thinning method, which is added to traditional IGBTs as a thin n-base and a low-doped field-stop layer, can significantly minimize Eoff [22]. In contrast to this feature, RC-IGBTs have a backside N+ doping area that reduces turn-off loss at the expense of substantial forward static losses. Therefore, a superjunction structure for RC-IGBTs. "double-side carrier storage layer" concept to improve conductance modulation in the drift zone and further reduce the Von compared with RC-IGBTs in [19-21].”
Point 3: Is there any expected difference when the simulation is expanded to 3D?
Response 3:
This manuscript gives simulation of a symmetric cell pitch in this new structure. Expansion of the 3D structure is also realized in this repetition of cell structure. In 3D structure, some inconsistency effect can be limited to a small extent by the layout. So in our design, main 3D structure is generally optimized in 2D simplified unit. Variations in characteristics and errors from 2D to 3D can be predicted.
Point 4: In Figure 5(c), it seems that the ref. [11] exhibited the snapback characteristic. However, the title of ref. [11] is "A novel snapback-free reverse conducting IGBT ~" and the ref. [11] showed the snapback-free characteristic in SH-RC-IGBT. The authors should revise/add some data/description.
Response 4:
Thank you very much to point out the ambiguous issues in our manuscript for snapback expression. The issue has been fixed in revised one.
In Figure 5(c), it seems that the ref. [11] exhibited the snapback characteristic in optimized IGBT with large forward voltage drop. Although ref. [11] gives "A novel snapback-free reverse conducting IGBT ~", the SH-RC-IGBT in ref. [11] obtains the snapback-free characteristic at expanse of voltage drop and difficulty of thyristor triggering. The description of Figure 5(c) in the manuscript introduces some ambiguities for RC-IGBT with Shockley Diode. Actually, in Figure 5(c), the reverse snapback-free is also demonstrated in ref.[11], and yet the corresponding reverse voltage drop is around 1V. If the voltage continues to decrease, the thickness of p2-base region should be reduced, which results in large forward voltage drop. The optimized design of p2-base plays key role in turning on parasitic thyristor. Hence, considering the comparison results between the proposed structure and the structure in ref.[11], we selected the comparison data under the same conditions to reflect the differences with reverse snapback characteristic.
Furthermore, in order to clarify this snapback-free effect of RC-IGBT with Shockley Diode, we revised the corresponding sentences in line 198-203 as
“In reference [11], the RC-IGBT with Schockley diode (SH-RC-IGBT) has a snapback-free at reverse voltage drop of 0.95V. Further lowering the reverse voltage drop in SH-RC-IGBT makes it challenging to activate the parasitic thyristor without snapback. Hence the RC-IGBT with Schockley diode (SH-RC-IGBT) has a snapback phenomenon even though it has the bigger τ with lower reverse voltage drop as shown is Figure 5(c), but it’s snapback free for the SJ-RC-IGBT, this is due to the electron can be injected into N-pillar through the boundary of the trench collector.”

Round 2
Reviewer 3 Report
I agree to the publication of this manuscript in Micromachines.